# Social Media Use and Alcohol Consumption: A 10-Year Systematic Review

**DOI:** 10.3390/ijerph191811796

**Published:** 2022-09-19

**Authors:** Saleem Alhabash, Sunyoung Park, Sandi Smith, Hanneke Hendriks, Yao Dong

**Affiliations:** 1Department of Advertising and Public Relations, College of Communication Arts and Sciences, Michigan State University, East Lansing, MI 48824, USA; 2Department of Communication Studies, California State University Long Beach, Long Beach, CA 90840, USA; 3Department of Communication, College of Communication Arts and Sciences, Michigan State University, East Lansing, MI 48824, USA; 4Hanneke Hendriks, Behavioral Science Institute, Radboud University, 6500 HE Nijmegen, The Netherlands

**Keywords:** social media, social networking sites (SNS), alcohol use, drinking, systematic review

## Abstract

Many studies have looked at the relationship between social media and alcohol consumption. There is a need for a comprehensive review that synthesizes the results of past research to systematically understand the relationship between social media use and alcohol consumption. The present systematic literature review synthesizes the findings from global social media and alcohol use studies (*n* = 206, 204 retained for analysis) between 2009 and 2019. Codes included type of study, methods, use of theory, and whether and how the relationship between social media and alcohol use was tested, among others. In addition to providing descriptive findings, the current study compared the findings across studies that primarily focused on advertising and marketing, self-generated user-generated content (UGC), other-generated UGC, social media uses and affordances, and a mixture of more than one type of content/focus. Most articles used quantitative methods (77.94%), which is followed by qualitative methods (15.20%), mixed methods (6.37%), and 0.49% that did not fit in any of the methods categories. Of the studies that tested the relationship between social media use and alcohol consumption, an overwhelming majority found that relationship to be positive (93.10%). The results of the present study provide a comprehensive understanding of past findings regarding social media and alcohol consumption and provide important future research suggestions.

## 1. Introduction

The rise of social media use, specifically among adolescents and young adults, has not only revolutionized human communication but has also magnified social and peer influences regarding social, political, and health-related issues [1]. One important area of influence is the relationship between social media use and alcohol consumption. Studies across different disciplines have examined exposure to user-generated (e.g., party pictures on Facebook, Instagram pictures displaying alcoholic drinks, Twitter posts about alcohol) and commercial (e.g., beer ads, interactive social media games) alcohol content on social media among young users and their associations with alcohol use and overuse [2,3,4]. Given the omnipresence of alcohol-related social media content, it is paramount to systematically understand the impact of such content on alcohol consumption.

The relationship between social media use and alcohol use has been examined from multiple disciplinary perspectives, including media and mass communication, psychology, health, and socio-technical system approaches, leveraging a diverse set of methodologies that cover behavioral and attitudinal research (self-report), psychophysiological and eye-tracking, as well as analysis of large-scale data sets using information system and computer science methodologies (e.g., machine learning and artificial intelligence techniques). The relationship between the two behaviors can be explained using social learning theory and social cognitive theory, both of which emphasize that humans learn or change their behaviors by observing others [5,6]. Seeing alcohol references on social media and interacting or engaging with alcohol-related content (e.g., like, comment, share) may, therefore, lead users to consume more alcohol themselves. Exposure to alcohol-related content and how it influences attitudes and behaviors could further be explained by the mere exposure effect [7,8], which articulates that repetitious exposure to any type of stimuli is bound to influence cognitive and affective attitudes as well as impact behavior; thus, favorable associations toward alcohol-related content are enhanced when alcohol-related stimuli appear repeatedly in one’s social media newsfeed.

Exposure to alcohol-related content is not the only affordance on social media platforms [9]. On social media, users passively receive information from others, but they can also actively generate their own content as well as enact online behaviors in response to content posted by others with this social network. The aspect of active engagement with content can also be explained, from a psychological perspective, using social learning and cognitive theories [10]. Both approaches emphasize that learning occurs by doing, and the more interactive the learning process is, the higher an individual’s self-efficacy in performing the behavior, especially when progress is rewarded. Therefore, engaging in posting about alcohol or engaging with others’ posts about alcohol, while holding the promise of reward (e.g., in the forms of likes) can facilitate an uptake of alcohol use and overuse. Such a relationship exists within a computer-mediated social structure; thus, the social norms approach is valuable here, as it postulates how individuals’ behaviors are shaped by their perceptions of the prevalence of that behavior among other people within their social/societal circle (varying reference groups) as well as by their perceptions of the acceptance of that behavior by others [5]. Seeing an alcohol post on social media may lead users to believe that others frequently engage in alcohol use and are relatively accepting of drinking (based on the often positive and social nature of these posts [11]).

The theoretical understanding of the relationship between alcohol use and social media use, outlined above, provides avenues for practical and health-related applications, risks, and concerns. Over a 10-year period (between 2012 and 2022), the average time spent on social media daily grew by over 63% to 147 min (almost 2.5 h) [12]. Social media networks continue to evolve in size, structure, and functions, thus rendering social media use as a habitual and ritualized aspect of users’ lives [13]. At the same time, the multitude of content types on social media continues to grow, and in relation to alcohol displays and discussions, there is an increasing volume and velocity of these conversations. Therefore, it is critical to examine the available academic literature as it relates to deciphering the relationship between use of social media and alcohol.

To date, there are two systematic literature reviews (SLR) [14,15] and one meta-analysis related to social media and alcohol use [16] with an additional meta-analysis related to social media use and risky behaviors among adolescents, including alcohol and substance use [17]. Despite their significant contributions, these studies are either broadly focused on overall risky behaviors including alcohol use, narrowly focused on exposure to commercial alcohol content, limited in methodological scope, or restricted in the extent of sources. The current SLR, guided by the Preferred Reporting Items for Systematic Reviews and Meta-Analyses (PRISMA) methods [18], provides a comprehensive systematic review of global social media and alcohol use studies between 2009 and 2019. It extends the scope of other reviews by including and comparing both user-generated and commercial alcohol-related content on social media. Furthermore, by analyzing both the results of these studies and their methodological and theoretical approaches, this SLR provides important directions for future research. Below, a more comprehensive overview of the previous reviews, definitions of key concepts, and research questions are presented.

### 1.1. Previous Reviews

As noted above, to date, there are two SLRs [14,15] and one meta-analysis [16] related to social media and alcohol use, with an additional meta-analysis related to social media use and risky behaviors among adolescents [17]. Each of these is discussed in turn.

The two SLRs relevant to this topic of inquiry [14,15] covered the relationship between general social media use and alcohol use [15] and the relationship between exposure to alcohol marketing and advertising, on one hand, and alcohol consumption, on the other [14]. The Lobstein et al. [15] review found that higher exposure to digital alcohol marketing was related to increased drinking behavior. In their “mini-review”, Groth and colleagues [14] summarized study findings regarding social media use and engagement in risk behaviors. Positive associations were also reported between social media use and several risk behaviors, which included but did not focus solely on alcohol consumption. The study, however, focuses specifically on studies related to alcohol-related social media use. This review solely focused on advertising content on social media.

Curtis and colleagues [16] conducted a meta-analysis to study the relation between alcohol-related social media engagement (e.g., posting, liking) and alcohol consumption and related problems. The study revealed a statistically significant relationship and moderate effect sizes between alcohol-related social media engagement and alcohol use and related problems. Although not solely focused on alcohol use, the meta-analysis of Vannucci et al. [17] focused on studies that examined adolescents’ use of social media within the context of risky behaviors, including substance use. The analysis identified 14 studies focused on substance use and found small-to-medium effect sizes linking social media use and substance use.

Thus, in sum, previous reviews and meta-analyses have significant implications and have made relevant contributions. However, these reviews are also either broadly focused on overall risky behaviors including alcohol use, narrowly focused on only exposure to commercial alcohol content, limited in methodological scope, or restricted in the extent of sources. This current SLR addresses all these gaps by including and comparing both UGC and commercial alcohol-related content on social media as well as studies focusing on linking general aspects of social media use to alcohol consumption. Additionally, the SLR approach followed in the current study expands the inclusion criteria to non-quantitative research approaches as well as studies published in a variety of publication venues. We also examine this phenomenon across different age groups, thus providing a more comprehensive overview of the existing literature. By analyzing not only the results of these studies but also their methodological and theoretical approaches, this SLR can provide directions for future research. Lastly, qualitative work, theoretically informed and designed studies, and peer influence are included in this review, because as Groth et al. [14] suggested, insights from studies of these types are essential to gain a more comprehensive understanding of the relationship between the two constructs.

### 1.2. Definitions of Key Concepts

In the current study, we aim to compare studies related to alcohol advertising and marketing, self-generated user-generated content (UGC), other-generated UGC, and social media use and affordances. Therefore, it is essential that we provide conceptual definitions of the major concepts relevant to this review. Social media refers to “Internet-based channels that allow users to opportunistically interact and selectively self-present, either in real-time or asynchronously, with both broad and narrow audiences who derive value from user-generated content and the perception of interaction with others” [19] (p. 50). On the other hand, alcohol use is defined as the frequency, volume, and patterns of alcohol consumption and intake [20].

Advertising and marketing content refers to paid, owned, or earned (i.e., shared) content distributed through social media with the intent to persuade someone to adopt or change their attitudes and/or behaviors [21]. As it relates to the current review, advertising studies are ones concerned with alcohol promotion by leading brands using a variety of online approaches to encourage engagement with a brand (e.g., interviews with celebrities, pictures and messaging accessories, notices of parties, brand-promoting Facebook pages, social media being used by local alcohol outlets, etc.) [14].

Our study also identified two types of UGC: self-generated and other-generated. Krumm, Davies, and Narayanaswami [22] define UGC as “content [that] comes from regular people who voluntarily contribute data, information, or media that then appears before others in a useful or entertaining way” on social media and other digital platforms. In the Curtis et al. [15] study, the authors operationalized UGC content on social by the inclusion of social media measures and constructs in the study (e.g., number of alcohol posts and density scores of alcohol images). Self-generated UGC refers to studies that focused on alcohol content created by participants themselves. Other-generated UGC refers to instances where participants in the study were exposed to UGC posts generated by others within their social media networks (e.g., a friend’s post). Thus, self-generated UGC deals with participants’ posting about alcohol, while other-generated UGC deals with exposure to posts by friends and acquaintances on social media platforms. It is worth mentioning, as noted later in the Findings section, that studies could potentially include different types of alcohol-related content, where the latter type also includes analysis of publicly available social media posts.

The last type of studies deals with studies that examined general social media use or entailed aspects of the platform’s affordances. This type includes studies that measured the amount of time spent on social media, the size of one’s network of friends on the platform, and specific uses of social media (e.g., use of alcohol-related hashtags).

### 1.3. Aims and Research Questions

This SLR has three overarching goals: **Aim (1)** To describe the prevalence, methods, and theories related to alcohol-related social media studies. **Aim (2)** To describe the findings regarding the relationship between social media and alcohol use. **Aim (3)** To compare studies focusing on user-generated and commercial alcohol content on social media. These aims are translated into three overarching research questions:**RQ1: Prevalence, methods, and theories:****RQ1a: Prevalence:** What is the prevalence of alcohol-related social media studies over time, by location, and by funding source?**RQ1b: Methodological approaches:** What are the methodological approaches (i.e., method, sample characteristics, and measurements) employed in alcohol-related social media studies?**RQ1c: Theoretical approaches:** What are the theoretical approaches employed in alcohol-related social media studies?**RQ2: Test of the relationship between social media and alcohol use:****RQ2a: Nature of relationship:** What was the nature of the tested relationship between social media and alcohol use (i.e., which was the predictor/IV and which was the criterion/DV)?**RQ2b: Findings:** What findings did studies report regarding this relationship?**RQ3: Comparing UGC and Commercial studies:**How do alcohol UGC and commercial content compare in terms of (a) prevalence, (b) methods, (c) sample characteristics, (d) measures, (e) theoretical approaches, and (f) their test of the relationship between social media and alcohol?

## 2. Materials and Methods

Per Fink [23], (p. 3 as cited in [24]) a systematic literature review is “a systematic, explicit, comprehensive, and reproducible method for identifying, evaluating, and synthesizing the existing body of completed and recorded work produced by researchers, scholars, and practitioners” (p. 17). SLRs analyze available literature on a particular topic to assess the nature of the literature, its quality and significance [24]. Xiao and Watson [25] identified four SLR that either describe a body of literature, textualize findings from the literature, extend past findings, and critique that body of work. The current SLR is descriptive in nature that uses a textual narrative synthesis approach relying on quantitative coding of studies included in the SLR [25]. The textual narrative synthesis descriptive approach is regarded as a more rigorous technique compared to narrative reviews, as it entails quantitative and qualitative assessment of major attributes and trends of an existing body of literature [25]. This approach is deemed acceptable for the current investigation, given that our objective is to provide a comprehensive overview of a broad area of research.

### 2.1. Search Strategy

We adopted the PRISMA approach for conducting SLR [18]. We used 10 different databases (CINAHL, Cochrane, Communication and Mass Media Complete, EMBASE, ProQuest, PsychInfo, PubMed, Science Direct, Scopus, and Web of Science) to search for global peer-reviewed journal articles and gray literature published between 2009 and 2019. Searches were conducted with keywords related to social media (e.g., “social media”, “social networking site”, “Facebook”, “Twitter”) and alcohol (“alcohol”, “drink”). Records were identified through an initial database search (*N* = 23,647). Trained researchers examined each database list of articles and performed initial screening of the records to ensure that the studies retrieved focus on both alcohol use and social media use. Upon initial screening and assessment of the relevance, we identified a total of 1119 studies as a combined sample across the different databases. The removal of duplicates yielded a sample of 467 studies (see Figure 1). After this, we performed an initial coding of the full text of each article to determine its eligibility to include 291 studies. Eligibility was assessed by ensuring that the study focused on both alcohol use and social media use. In some cases, specifically in cases where big data sets of social media posts were used, the emphasis was on social media content that referenced alcohol. Thus, our inclusion/exclusion criteria was mainly focused on studies that (1) tested the relationship between social media use variables and alcohol use variables, (2) examined the phenomenon of social media use in association with alcohol consumption, and/or (3) studies that examined alcohol-related social media content, either as a standalone phenomenon or linked to other forms of data that built a connection between alcohol-related social media content and drinking patterns and prevalence among different populations. We then excluded 85 studies that were irrelevant to our search topics, such as alcohol studies using social media as a channel to recruit participants, errata/commentary, and interventions. This resulted in a total of 206 studies that were included in the analysis.

### 2.2. Coding Process and Codebook

Upon thoroughly screening articles (*n* = 291), a team of four independent coders pretested the coding scheme with five articles. The four coders met and resolved differences, which has yielded a revised version of the codebook. The four coders then independently coded 10% of the sample, which resulted in an 89% average agreement among the coders. During this coding phase, coding categories with percentage agreement lower than 75% we assessed closely, and the coders refined their interpretation of these categories to ensure reliable coding of the entire sample. The articles coded for inter-rater reliability were then integrated into the larger data set by recording each category by majority rule. In a minority of cases, where agreement was split among the coders, one of the coders re-examined the article and recoded that category based on the refined understanding of the codebook.

We developed an extensive codebook that examines the type of study, method, sample characteristics, (predictors of) alcohol consumption measures, social media platform, and social media content. For a full description of the codebook, refer to Table 1 (left-most column).

## 3. Results

### 3.1. RQ1: Prevalence, Methods, and Theories

#### 3.1.1. Prevalence (RQ1a)

Of the 206 studies, two were excluded as they did not fit any of the content type categories; thus, 204 studies were retained for all subsequent analyses. Most studies (*n* = 108; 52.94%) included a mixture of advertising/marketing content, self- and other-generated UGC, and social media use and affordances measures, which were followed by 51 studies (25.00%) that focused solely on self-generated UGC, 22 studies (10.78%) that focused on other-generated UGC, 14 studies (6.86%) that focused on social media use and affordances, and nine studies (4.12%) that focused exclusively on alcohol advertising and marketing (see Figure 2), χ^2^(4) = 164.28, *p* < 0.001.

Most coded articles were classified as research articles published in peer-reviewed journals (*n* = 167, 81.86%), which were followed by research abstracts (*n* = 15, 7.35%), conference papers (*n* = 10, 4.90%), graduate theses and dissertations (*n* = 7, 3.43%), and five book chapters (2.45%), respectively, χ^2^(4) = 489.33, *p* < 0.001 (see Figure 3). Most articles (*n* = 198, 97.06%) reported findings from a single study, with only four articles that reported findings from two studies (1.96%); the remaining two studies comprised a conceptual piece published in the Youth Drinking Cultures in a Digital World book [26] and an opinion piece [27] that articulated the relationship between the two constructs on a conceptual basis, χ^2^(2) = 372.82, *p* < 0.001.

A total of 12 studies (5.88%) did not specify a geographic location for the conduct of the study. Over half of the studies (*n* = 105, 51.47%) were situated in the United States, with the remaining 87 studies distributed across the following countries, respectively: Australia (*n* = 19, 9.31%), the United Kingdom (*n* = 16, 7.84%), New Zealand (*n* = 12, 5.88%), Belgium (*n* = 7, 3.43%), Norway (*n* = 5, 2.45%), the Netherlands (*n* = 4, 1.96%), Mexico (*n* = 3, 1.47%), two studies each (0.98%) from Canada, Sweden, and Switzerland, and a single study each (0.49%) from Italy, Korea, and Thailand. Additionally, 12 studies (5.88%) focused on multiple countries, with five of those focusing on Australia and India, one study conducted in Belgium and the United States, one study in England and Australia, one study in South Korea and the United States, one study in the United Kingdom, Ireland, and Australia, one study from the Netherlands and Germany, and two global studies with multiple countries. Figure 4 shows the geographical distribution of the countries where the research was conducted by studies included in this review. For comparative analyses, the country variable was recoded into 1 = United States Only and 0 = Non-US (including cross-country studies).

Over half of the studies (*n* = 106, 51.45%) reviewed disclosed support from funding entities, including governmental agencies, foundations, and other sources.

#### 3.1.2. Methodological Approaches (RQ1b)

In terms of research methodology, the majority of the articles (*n* = 159, 77.94%) used quantitative methods, followed by 31 articles that used qualitative methods (15.20%), 13 articles that used mixed methods (6.37%), and 1 article (0.49%) that did not follow a particular research methodology, χ^2^(3) = 313.88, *p* < 0.001. For the studies that used quantitative methods (including those leveraging mixed-method approaches), 79 (48.77%) used survey methodology, followed by content analysis (*n* = 34, 20.99%), big-data mining (*n* = 22, 13.58%), experimental research (*n* = 14, 8.64%), and multiple quantitative methods (often surveys with content analysis or big data mining; *n* = 14, 8.02%), χ^2^(4) = 92.51, *p* < 0.001. Of the 35 studies that used qualitative methods, 14 used focus groups (40.00%), four used interviews (11.43%), five used textual analysis (14.29%), eight used multiple qualitative methods (22.86%), and four articles that used qualitative methods not listed in our coding scheme (11.43%), χ^2^(4) = 10.29, *p* = 0.04. Of the articles leveraging quantitative methodologies and mixed methodology, 71 used cross-sectional designs, while only 21 used longitudinal research designs, χ^2^(1) = 27.17, *p* < 0.001, thus indicating a much higher prevalence of cross-sectional than longitudinal designs (Figure 5).

About half of the studies (*n* = 97, 47.55%) were conducted online, followed by non-human subjects (*n* = 33, 16.18%), field studies (*n* = 31, 15.20%), and studies conducted in a lab or research facility (*n* = 26, 12.75%). An additional 17 studies (8.33%) did not specify the context of the research, χ^2^(4) = 100.51, *p* < 0.001 (Figure 6).

In terms of the age ranges of participants, we classified studies based on the minimum legal age of 21 to purchase of alcoholic beverages in the United States. Based on this, over one-fifth of the sample (*n* = 45, 22.06%) recruited participants below the age of 21, 61 studies (29.90%) recruited adult samples (over 21 years old), and 16 studies (7.84%) recruited samples of underage youth and adults. The remaining 82 studies (40.20%) did not indicate the age of participants, bearing in mind that some of these studies are non-human subjects studies, χ^2^(3) = 45.53, *p* < 0.001 (see Figure 7). Roughly six of 10 of the reviewed studies (*n* = 117, 57.35%) included a reference in the method section related to the sample’s gender distribution, χ^2^(1) = 4.41, *p* = 0.04.

**Alcohol Measures**. A minority of the studies (*n* = 25, 12.25%) exclusively recruited drinkers, χ^2^(1) = 116.26, *p* < 0.001. Less than half of the studies (*n* = 97, 47.55%) relied on self-report measures of alcohol use, and 106 studies (51.96%) did not explicitly indicate the way alcohol use was measured, with only one study using unobtrusive measures of alcohol use such as blood alcohol concentration (BAC) and breathalyzer, χ^2^(2) = 99.62, *p* < 0.001. Of the 204 studies, 83 (40.69%) measured alcohol use within a specific timeframe, χ^2^(1) = 7.08, *p* < 0.01. Thirty-one studies (15.20%) specifically used the alcohol use disorders identification test (AUDIT) to index alcohol use, with 47 additional studies (23.04%) that relied on alternative clinical and standardized measures of alcohol such as the Daily Drinking Questionnaire (DDQ), NIDA Modified ASSIST substance use screener, Quantity Frequency Index, and the timeline follow back (TFLB) method. Roughly four in 10 studies (*n* = 77, 37.38%) measured perceptions of alcohol use among peers and friends were measured, χ^2^(1) = 12.26, *p* < 0.001. About one-fifth of the studies reviewed (*n* = 40, 19.60%) included a social norms measure related to alcohol use, χ^2^(1) = 75.27, *p* < 0.001. Seventy-six studies (37.25%) included a reference of excessive drinking, χ^2^(1) = 13.26, *p* < 0.001. An extreme majority of studies did not reference short- and long-term risks associated with drinking (*n* = 195, 95.59%), χ^2^(1) = 169.59, *p* < 0.001, where nine studies included references to negative consequences of alcohol use, which included a single study focused on long-term consequences, seven focused on short-term consequences, and a single study that focused on both long- and short-term consequences. A minority of the studies reviewed (*n* = 29, 14.22%) measured the use of other drugs concurrently with alcohol use, χ^2^(1) = 104.49, *p* < 0.001. Finally, only four studies focused on celebration drinking, χ^2^(1) = 188.31, *p* < 0.001.

**Social Media Measures**. The top social media platforms in terms of use [28] were also represented regarding alcohol-related social media studies. Facebook was the most popular standalone platform as the context of research, followed by Twitter, Instagram, and YouTube, respectively. Less than one-fifth of the studies (*n* = 38, 18.63%) referenced general social media use in the studies, and a similar portion of the sample (*n* = 39, 19.12%) included multiple social media platforms for investigation, and 10 studies (4.90%) investigated other platforms such as Myspace and Reddit, χ^2^(6) = 123.08, *p* < 0.001 (see Figure 8). Of the 204 studies, 171 (83.33%) included a measure of alcohol-related social media use, χ^2^(1) = 93.35, *p* < 0.001. Thirty-seven (18.14%) of the studies included a measure of time spent on social media, χ^2^(1) = 82.84, *p* < 0.001. Additionally, 20 studies (9.80%) measured the number of social media friends, χ^2^(1) = 131.84, *p* < 0.001.

#### 3.1.3. Theoretical Approaches (RQ1c)

Over two-thirds (*n* = 139, 67.16%) of the coded articles did not include any theoretical framework. The 67 studies (32.84%) using theories leveraged 39 different theories, theoretical models, frameworks, and concepts, χ^2^(1) = 24.02, *p* < 0.001 (see Table 1 for a full list of theories used). Studies referenced more than one theory, model, framework, or theoretical concept. Multiple studies referenced the same theory. We further coded the theories into disciplines. Of the 39 theories used across the 67 studies, 31 were psychological, 3 were sociological, 2 were criminological, 2 were critical, and 1 theory was newly developed.

As noted in Table 1, most of the theories had a single frequency among the studies. A few theories, specifically those within the psychological tradition, had multiple occurrences across the literature. Specifically related to psychological theories, the most frequently used theoretical approaches were the social norms approach, theory of reasoned action and the theory of planned behavior (combined frequency), and social learning theory and social cognitive theory (combined frequency), respectively. Of the other theoretical disciplines, the only theory with multiple occurrences was grounded theory within the critical theoretical approach.

### 3.2. RQ2: Relationship between Social Media and Alcohol Use

#### 3.2.1. Nature of Relationship (RQ2a)

To test RQ2a we coded alcohol use and social media use in each study where the relationship was formally tested as predictor/IV, criterion/DV, moderator, mediator, or control variable. For convenience’s sake, a three-category variable was created for each coded variable where 1 = predictor/IV, 2 = criterion/DV, and 3 = Other, which included cases where either variable was a mediator, moderator, or control variable. Additionally, we added instances where a correlational (often bivariate) relationship was tested; thus, each variable was regarded as both predictor and criterion. Of 96 studies that measured alcohol use, an overwhelming majority of 83 articles (86.46%) measured alcohol use as a criterion or dependent variable, only six (6.25%) articles included it as a predictor or independent variable, and seven others (7.29%) included it either as a mediator, moderator, control, or both predictor and criterion, χ^2^(2) = 121.94, *p* < 0.001. On the other hand, of the 111 articles that measured social media use, 89 articles (80.18%) measured it as a predictor or IV, 16 articles (14.41%) measured it as criterion/DV, and six articles (5.41%) measured it as a different type of variable, χ^2^(2) = 110.97, *p* < 0.001. Thus, most studies measured social media use as a predictor/IV and alcohol use as a criterion/DV. Exceptions included studies where the relationship was reciprocal, alcohol use was a control variable, or either construct regarded as a mediator or moderator.

#### 3.2.2. Study Findings (RQ2b)

A total of 163 examined the relationship between social media and alcohol use (only quantitative studies and mixed methods with quantitative approaches). With that in mind, of the 162 cases viable for coding here, 87 studies (53.70%) formally tested the relationship between the two constructs, and the remaining 75 studies did not conduct a formal test of that relationship. Of the 87 studies that formally tested the relationship, an overwhelming majority found that relationship to be positive (*n* = 81, 93.10%), five studies (5.74%) did not find a relationship between the two variables, and a single study (1.15%) found the relationship to be negative in direction, χ^2^(2) = 140.14, *p* < 0.001.

### 3.3. RQ3: Comparing Social Media Content and Uses Types

The study’s second set of research questions dealt with examining differences across study types in relation to their focus on social media content and uses types (SMCU types) examined in the study. Descriptive statistics (frequencies and percentages) along with FET test value and *p*-values are presented in Table 2. The following will only highlight the significant findings of the FET for the examined coding variables.

#### 3.3.1. Prevalence of Studies (RQ3a)

Studies did not vary significantly in terms of publication venue, with the majority of studies across social media content and use types appearing in peer-reviewed journals. Regarding the country where the study was conducted (sample selection), all SMCU types studied, except mixed content types’ studies, were more heavily focused on the United States, where the mixed content types’ studies had a greater prevalence of non-U.S. samples. Studies across SMCU types were somewhat evenly split in terms of funding.

#### 3.3.2. Research Methodology (RQ3b)

Study types were uniformly distributed in terms of the research methodology employed in the study (no significant differences), with most studies using quantitative methods. However, for studies employing quantitative methodology, the differences were significant in that advertising/marketing studies were most experimental and content-analytic; self-UGC studies were mostly survey-based, followed by content analytic, big-data and multiple-method studies (no experimental studies); other-UGC studies were mostly survey-based, followed by experimental, content analytic, big data, and multiple methods; SM Use and Affordances studies were mostly survey-based with one study using multiple quantitative methods; about four in 10 of the SMCU types studies were survey-based, followed by a quarter that used content analysis, one in five using big data, and less than 10% using each experimental and multiple methods. No significant differences were detected for qualitative study categories, research design (mostly cross-sectional), and distribution method (mostly online).

#### 3.3.3. Sample Characteristics (RQ3c)

Differences between SMCU types in terms of participant age were marginally significant. Although a minority of studies across SMCU types focused on participants younger than 21 years old, the different types slightly varied: one advertising/marketing study, one-third of self-UGC studies, over one in five of other-UGC studies, and less than one in five of mixed SMCU types studies focused on individuals under the age of 21. SMCU types also varied in terms of reporting the gender distribution of participants, where the majority of studies in all types, except mixed-SMCU types, reported the gender distribution of their samples; over half of mixed SMCU types studies did not report the gender distribution of the study sample. Finally, the SMCU types did not differ in terms of focusing solely on alcohol users (drinkers), where most studies recruited both drinkers and non-drinkers.

#### 3.3.4. Alcohol Use and Social Media Use Measures (RQ3d)

Studies across the different SMCU types mostly relied on self-report measures of alcohol use. Studies varied in terms of measuring alcohol in a specified timeframe (e.g., in the past week, month, year, etc.), where advertising/marketing, other-UGC, and mixed SMCU types studies had a higher frequency of studies that did not include a time-based measure of alcohol use (between two-thirds and seven in 10), while most studies in the self-UGC and SM use and affordances categories included a time-based measure of alcohol use. With regard to use of clinical alcohol use measures (e.g., AUDIT), self-UGC studies had the highest prevalence of studies that used AUDIT and other clinical measures (over half), where most other studies did not specify a clinical measure of alcohol use. Advertising/marketing studies and mixed SMCU types studies did not include a measure of drinking among reference groups and/or friends, whereas, about half the studies coded as self-UGC, other-UGC, and SM use and affordances included such measures. Most studies in all SMCU types categories (over 85%), except SM use and affordances, did not integrate measures of other drugs in the study, where over four in 10 SM use and affordances studies included such measures. SMCU types did not vary in terms of focus on social norms, excessive drinking, negative consequences of drinking, and celebration drinking.

Regarding social media use measures, the different SMCU types varied significantly in terms of the social media focus in the study and inclusion of social media use frequency (time) but not in terms of including a measure of the number of social media friends. For the social media context of the study, Facebook was the most frequent content for advertising/marketing and self-UGC studies, while other SMCU types were more distributed across the different platforms. In contrast to other SMCU types that did not include a measure of social media use frequency (time), over half of SM use and affordances studies included a social media use time measure.

#### 3.3.5. Theory (RQ3e)

All SMCU types, except other-UGC and SM uses and affordances, had a higher frequency of studies without a theoretical framework. Over half of other-UGC and half of SM uses and affordances studies included a theory.

#### 3.3.6. Relationship Testing (RQ3f)

We observed uniformity across SMCU types in the designation of alcohol use as a criterion or DV and social media use as a predictor or IV. SMCU types varied significantly regarding the formal testing of the relationship between social media use and alcohol consumption: nearly six in 10 advertising/marketing and mixed SMCU types studies did not test the relationship, while most self-UGC (nearly three-quarters), other-UGC (over half), and SM uses and affordances (over eight in 10) tested it. When that relationship was tested in the studies across SMCU types, it was found to be positive in nature.

**Table 2 ijerph-19-11796-t002:** Descriptive and Chi-Square results for coded variables and by study type.

Variable	All	Study Type					χ^2^ (Study Type)
		AD	Self-UGC	Other-UGC	SM Use	Mixed	
	*N* (%)	*N* (%)	*N* (%)	*N* (%)	*N* (%)	* N * (%)	
**Publication venue**	***n* = 204**	***n *= ** **9**	***n *= ** **51**	***n *= ** **22**	***n *= ** **14**	***n *= ** **108**	χ^2^(16) = 18.60, *ns*
Journal	167 (81.86%)	9 (100%)	39 (76.47%)	15 (68.18%)	11 (78.57%)	93 (86.11%)	
Book chapter	5 (2.45%)	0 (0.00%)	3 (5.88%)	0 (0.00%)	1 (7.14%)	1 (.93%)	
Conf. paper	10 (4.90%)	0 (0.00%)	2 (3.92%)	1 (4.55%)	1 (7.14%)	6 (5.56%)	
Abstract	15 (7.35%)	0 (0.00%)	6 (11.76%)	4 (18.18%)	0 (0.00%)	5 (4.63%)	
Diss./thesis	7 (3.43%)	0 (0.00%)	1 (1.96%)	2 (0.909%)	1 (7.14%)	3 (2.78%)	
** *Chi-Square (All)* **	χ^2^(4) = 489.33, *p* < 0.001						
**Country**	***n* = 192**	***n *= ** **8**	***n *= ** **49**	** *n* ** **= 22**	** *n* ** **= 13**	** *n* ** **= 100**	χ^2^(4) = 11.94, *p* = 0.02
Non-U.S.	87 (45.31%)	3 (37.50%)	17 (34.69%)	6 (27.27%)	4 (30.77%)	57 (57.00%)	
U.S.	105 (54.69%)	5 (62.50%)	32 (65.31%)	16 (72.73%)	9 (69.23%)	43 (43.00%)	
** *Chi-Square (All)* **	χ^2^(1) = 1.69, *ns*						
**Funding disclosure**	***n* = 204**	***n *= ** **9**	***n *= ** **51**	***n *= ** **22**	***n *= ** **14**	***n *= ** **108**	χ^2^(4) = 3.45, *ns*
Yes	106 (51.96%)	4 (44.44%)	19 (37.25%)	12 (54.55%)	7 (50.00%)	56 (51.85%)	
No	98 (48.04%)	5 (55.56%)	32 (62.75%)	10 (45.45%)	7 (50.00%)	52 (49.06%)	
** *Chi-Square (All)* **	χ^2^(1) = 0.31, *ns*						
**Research Method**	***n *= 204**	***n *= ** **9**	***n *= ** **51**	***n *= ** **22**	***n *= ** **14**	***n *= ** **108**	χ^2^(12) = 3.53, *ns*
Quantitative	159 (77.94)	8 (88.89%)	38 (74.51%)	19 (86.36%)	11 (78.57%)	83 (76.85%)	
Qualitative	31 (15.20%)	1 (11.11%)	10 (19.61%)	2 (9.09)	2 (14.29%)	16 (14.81%)	
Mixed Methods	13 (6.37%)	0 (0.00%)	3 (5.88%)	1 (4.55%)	1 (7.14%)	8 (7.41%)	
Other	1 (0.03%)	0 (0.00%)	0 (0.00%)	0 (0.00%)	0 (0.00%)	1 (0.93%)	
** *Chi-Square (All)* **	χ^2^(3) = 313.88, *p* < 0.001						
** *Quantitative* **	***n* = 162**	***n *= ** **8**	***n *= ** **38**	***n *= ** **20**	***n *= ** **11**	***n *= ** **85**	χ^2^(16) = 47.26, *p* < 0.001
Survey	79 (48.77%)	1 (12.50%)	22 (57.89%)	11 (55.00%)	9 (81.82%)	36 (42.35%)	
Experiment	14 (8.64%)	4 (50.00%)	0 (0.00%)	4 (20.00%)	0 (0.00%)	6 (7.06%)	
Content analysis	34 (20.99%)	3 (37.50%)	6 (15.79%)	2 (10.00%)	1 (9.09%)	22 (25.88%)	
Big data	22 (13.58%)	0 (0.00%)	3 (7.89%)	2 (10.00%)	0 (0.00%)	17 (20.00%)	
Multiple	13 (8.02%)	0 (0.00%)	7 (18.42%)	1 (5.00%)	1 (9.09%)	5 (4.71%)	
** *Chi-Square (All)* **	χ^2^(4) = 92.51, *p* < 0.001						
** *Qualitative* **	***n* = 35**	***n *= ** **1**	***n *= ** **10**	** *N =* ** **3**	***n *= ** **2**	***n *= ** **19**	χ^2^(16) = 20.88, *ns*
Focus groups	14 (40.00%)	0 (0.00%)	5 (50.00%)	3 (100.00%)	0 (0.00%)	6 (31.58%)	
Interviews	4 (11.43%)	0 (0.00%)	3 (30.00%)	0 (0.00%)	0 (0.00%)	1 (5.26%)	
Textual analysis	5 (14.29%)	0 (0.00%)	0 (0.00%)	0 (0.00%)	0 (0.00%)	5 (26.32%)	
Multiple	8 (22.86%)	1 (100.00%)	1 (10.00%	0 (0.00%)	1 (50.00%)	5 (26.32%)	
Other	4 (11.43%)	0 (0.00%)	1 (10.00%)	0 (0.00%)	1 (50.00%)	2 (10.53%)	
** *Chi-Square (All)* **	χ^2^(4) = 10.29, *p* = 0.04						
**Research Design**	***n* = 92**	***n *= ** **1**	***n *= ** **29**	***n *= ** **12**	** *n* ** **= 10**	***n *= ** **40**	χ^2^(4) = 2.13, *ns*
Cross-sectional	71 (77.17%)	1 (100.00%)	20 (68.97%)	9 (75.50%)	8 (80.00%)	33 (82.50%)	
Longitudinal	21 (22.83%)	0 (0.00%)	9 (31.03%)	3 (25.00%)	2 (20.00%)	7 (17.50%)	
** *Chi-Square (All)* **	χ^2^(1) = 27.17, *p* < 0.001						
**Distrib. Method**	***n* = 204**	***n *= ** **9**	***n *= ** **51**	***n *= ** **22**	***n *= ** **14**	***n *= ** **108**	χ^2^(16) = 13.01, *ns*
Lab/Res. Facility	26 (9.80%)	2 (22.22%)	8 (15.69%)	3 (13.64%)	1 (7.14%)	12 (11.11%)	
Field	31 (15.20%)	0 (0.00%)	5 (9.80%)	2 (9.09%)	2 (14.29%)	11 (10.19%)	
Online	97 (47.55%)	6 (66.67%)	29 (56.86%)	10 (45.45%)	7 (50.00%)	49 (45.27%)	
Non-human sub.	33 (16.18%)	1 (11.11%)	4 (7.84%)	3 (13.64%)	1 (7.14%)	24 (22.22%)	
Unspecified	17 (8.33%)	0 (0.00%)	5 (9.80%)	4 (18.18%)	3 (21.43%)	12 (11.11%)	
** *Chi-Square (All)* **	χ^2^(4) = 100.51, *p* < 0.001						
**Participant Age**	***n* = 204**	***n *= ** **9**	***n *= ** **51**	***n *= ** **22**	***n *= ** **14**	***n *= ** **108**	χ^2^(12) = 19.16, *p* = 0.085
Below 21	45 (22.06%)	1 (11.11%)	16 (31.37%)	5 (22.73%)	5 (35.71%)	18 (16.67%)	
Adults	61 (29.90%)	3 (33.33%)	20 (39.22%)	5 (22.73%)	5 (5.71%)	28 (25.93%)	
Mixed	16 (7.84%)	2 (22.22%)	4 (7.84%)	1 (4.55%)	0 (0.00%)	9 (8.33%)	
Unspecified	82 (40.20%)	3 (33.33%)	11 (21.57%)	11 (50.00%)	4 (28.57%)	53 (49.07)	
** *Chi-Square (All)* **	χ^2^(4) = 45.53, *p* < 0.001						
**Gender Distrib.**	***n* = 204**	***n *= ** **9**	***n *= ** **51**	***n *= ** **22**	***n *= ** **14**	***n *= ** **108**	χ^2^(4) = 14.01, *p* = 0.007
Reported	117 (57.35%)	6 (66.67%)	35 (68.63%)	16 (72.73%)	11 (78.57%)	49 (45.37%)	
Not reported	87 (42.65%)	3 (33.33%)	16 (31.37%)	6 (27.27%)	3 (21.43%)	59 (54.63%)	
** *Chi-Square (All)* **	χ^2^(1) = 4.41, *p* = 0.036						
**Part. Drinker Status**	***n* = 204**	***n *= ** **9**	***n *= ** **51**	***n *= ** **22**	***n *= ** **14**	***n *= ** **108**	χ^2^(4) = 7.04, *ns*
Drinkers and non-drinkers	179 (87.75%)	7 (77.78%)	40 (78.43%)	20 (90.91%)	13 (92.86%)	99 (91.67%)	
Drinkers only	25 (12.25%)	2 (22.2250	11 (21.57%)	2 (9.09%)	1 (7.14%)	9 (8.33%)	
** *Chi-Square (All)* **	χ^2^(1) = 116.26, *p* < 0.001						
**Alcohol measure**	***n* = 204**	***n *= ** **9**	***n *= ** **51**	***n *= ** **22**	***n *= ** **14**	***n *= ** **108**	χ^2^(8) = 11.88, *ns*
Self-report	97 (47.55%)	4 (44.44%)	33 (64.71%)	9 (40.91%)	9 (64.29%)	42 (38.89%)	
Unobtrusive	1 (4.90%)	0 (0.00%)	0 (0.00%)	0 (0.00%)	0 (0.00%)	1 (0.93%)	
Unspecified	106 (51.96%)	5 (55.56%)	18 (35.29%)	13 (59.09%)	5 (35.71%)	65 (50.19%)	
** *Chi-Square (All)* **	χ^2^(1) = 99.62, *p* < 0.001						
**Alc. Use Timeframe**	***n* = 204**	***n *= ** **9**	***n *= ** **51**	***n *= ** **22**	***n *= ** **14**	***n *= ** **108**	χ^2^(4) = 17.61, *p* = 0.001
No	121 (59.31%)	6 (66.67%)	20 (39.22%)	14 (63.64%)	5 (35.71%)	76 (70.37%)	
Yes	83 (40.69%)	3 (33.33%)	31 (60.78%)	8 (36.36%)	9 (64.29%)	32 (29.63%)	
** *Chi-Square (All)* **	χ^2^(1) = 7.08, *p* = 0.008						
**Clinical Alcohol Measure**	***n* = 204**	***n *= ** **9**	***n *= ** **51**	***n *= ** **22**	***n *= ** **14**	***n *= ** **108**	χ^2^(8) = 16.21, *p* = 0.004
AUDIT	31 (15.20%)	1 (11.11%)	14 (27.45%)	1 (4.55%)	2 (14.29%)	13 (12.04%)	
Other Clin. Meas.	47 (23.04%)	3 (33.33%)	15 (29.41%)	7 (31.91%)	4 (28.57%)	18 (16.67%)	
Unspecified	126 (61.76%)	5 (55.56%)	22 (43.14%)	14 (63.64%)	8 (57.14%)	77 (71.30%)	
** *Chi-Square (All)* **	χ^2^(1) = 76.09, *p* < 0.001						
**Reference Group—Peers/Friends**	***n* = 204**	***n *= ** **9**	***n *= ** **51**	***n *= ** **22**	***n *= ** **14**	***n *= ** **108**	χ^2^(4) = 12.04, *p* = 0.017
No	127 (62.25%)	8 (88.89%)	26 (50.98%)	10 (45.45%)	7 (50.00%)	76 (70.37%)	
Yes	77 (37.75%)	1 (11.11%)	25 (49.02%)	12 (54.55%)	7 (50.00%)	32 (29.63%)	
** *Chi-Square (All)* **	χ^2^(1) = 12.26, *p* < 0.001						
**Social Norms**	***n* = 204**	***n *= ** **9**	***n *= ** **51**	***n *= ** **22**	***n *= ** **14**	***n *= ** **108**	χ^2^(4) = 3.02, *ns*
Not measured	164 (80.39%)	8 (88.89%)	43 (84.31%)	15 (68.18%)	11 (78.57%)	87 (80.56%)	
Measured	40 (19.62%)	1 (11.11%)	9 (15.69%)	7 (31.82%)	3 (21.43%)	21 (19.44%)	
** *Chi-Square (All)* **	χ^2^(1) = 75.37, *p* < 0.001						
**Excessive drinking**	***n* = 204**	***n *= ** **9**	***n *= ** **51**	***n *= ** **22**	***n *= ** **14**	***n *= ** **108**	χ^2^(4) = 5.74, *ns*
Not measured	128 (62.75%)	7 (77.78%	26 (50.98%)	13 (59.09%)	8 (57.14%)	74 (68.52%)	
Measured	76 (37.25%)	2 (22.22%)	25 (49.02%)	9 (40.91%)	6 (42.86%)	34 (31.48%)	
** *Chi-Square (All)* **	χ^2^(1) = 13.26, *p* < 0.001						
**Risks**	***n* = 204**	***n *= ** **9**	***n *= ** **51**	***n *= ** **22**	***n *= ** **14**	***n *= ** **108**	χ^2^(4) = 1.05, *ns*
Not measured	195 (95.59%)	9 (10.00%)	48 (94.12%)	21 (95.45%)	13 (92.86%)	104 (96.30%)	
Measured	9 (4.41%)	0 (0.00%)	3 (5.88%)	1 (4.55%)	1 (7.14%)	4 (3.70%)	
** *Chi-Square (All)* **	χ^2^(1) = 169.59, *p* < 0.001						
**Use of Other Drugs**	***n* = 204**	***n *= ** **9**	***n *= ** **51**	***n *= ** **22**	***n *= ** **14**	***n *= ** **108**	χ^2^(4) = 10.39, *p* = 0.034
Not measured	175 (85.78%)	8 (88.89%)	44 (86.27%)	20 (90.91%)	8 (57.14%)	95 (87.96%)	
Measured	29 (14.22%)	1 (11.11%)	7 (13.73%)	2 (9.09%)	6 (42.86%)	13 (12.04%)	
** *Chi-Square (All)* **	χ^2^(1) = 104.49, *p* < 0.001						
**Celebration drink.**	***n* = 204**	***n *= ** **9**	***n *= ** **51**	***n *= ** **22**	***n *= ** **14**	***n *= ** **108**	χ^2^(4) = 2.85, *ns*
Not measured	200 (98.04%)	9 (100.00%)	49 (96.08%)	21 (95.45%)	14 (100%)	107 (99.07%)	
Measured	4 (1.96%)	0 (0.00%)	2 (3.92%)	1 (4.55%)	0 (0.00%)	1 (0.93%)	
** *Chi-Square (All)* **	χ^2^(1) = 188.31, *p* < 0.001						
**Social Media**	***n* = 204**	***n *= ** **9**	***n *= ** **51**	***n *= ** **22**	***n *= ** **14**	***n *= ** **108**	χ^2^(24) = 49.47, *p* = 0.002
General SM	38 (18.63%)	0 (0.00%)	7 (13.73%)	5 (22.73%)	7 (50.00%)	19 (17.59%)	
Facebook	76 (37.25%)	6 (66.67%)	30 (58.82%)	8 (36.36%)	2 (14.29%)	30 (27.78%)	
Twitter	10 (4.90%)	1 (11.11%	2 (3.92%)	0 (0.00%)	2 (14.29%)	5 (4.63%)	
Instagram	23 (11.27%)	0 (0.00%)	0 (0.00%)	4 (18.18%)	0 (0.00%)	19 (17.59%)	
YouTube	9 (4.41%)	0 (0.00%)	0 (0.00%)	0 (0.00%)	0 (0.00%)	9 (8.33%)	
Multi. platforms	39 (19.12%)	2 (22.22%)	10 (19.61%)	4 (18.18%)	2 (14.29%)	21 (19.44%)	
Other platforms	9 (4.41%)	0 (0.00%)	2 (3.92%)	1 (4.55%)	1 (7.14%)	5 (4.63%)	
** *Chi-Square (All)* **	χ^2^(6) = 123.08, *p* < 0.001						
**Time spent on SM**	***n* = 204**	***n *= ** **9**	***n *= ** **51**	***n *= ** **22**	***n *= ** **14**	***n *= ** **108**	χ^2^(4) = 17.56, *p* = 0.002
Not measured	167 (81.86%)	7 (77.78%)	46 (90.20%)	17 (77.27%)	6 (42.86%)	91 (84.26%)	
Measured	37 (18.14%)	2 (22.22%)	5 (9.80%)	5 (22.73%)	8 (57.14%)	17 (15.74%)	
** *Chi-Square (All)* **	χ^2^(1) = 82.84, *p* < 0.001						
**Number of friends**	***n* = 204**	***n *= ** **9**	***n *= ** **51**	***n *= ** **22**	***n *= ** **14**	***n *= ** **108**	χ^2^(4) = 0.53, *ns*
Not measured	184 (90.20%)	8 (88.89%)	46 (90.20%)	19 (86.36%)	13 (92.86%)	98 (90.74%)	
Measured	20 (9.80%)	1 (11.11%)	5 (9.80%)	3 (13.64%)	1 (7.14%)	10 (9.26%)	
** *Chi-Square (All)* **	χ^2^(1) = 131.84, *p* < 0.001						
**Theory**	***n* = 204**	***n *= ** **9**	***n *= ** **51**	***n *= ** **22**	***n *= ** **14**	***n *= ** **108**	χ^2^(4) = 12.65, *p* = 0.013
No theory used	137 (67.16%)	6 (66.67%)	33 (64.71%)	9 (40.91%)	7 (50.00%)	82 (75.93%)	
Theory used	67 (32.84%)	3 (33.33%)	18 (35.29%)	13 (59.09%)	7 (50.00%)	26 (24.07%)	
** *Chi-Square (All)* **	χ^2^(1) = 24.02, *p* < 0.001						
**Alc. Use Measure**	***n* = 96**	***n *= ** **4**	***n *= ** **27**	***n *= ** **12**	***n *= ** **11**	***n *= ** **42**	χ^2^(8) = 10.00, *ns*
Predictor	6 (6.25%)	0 (0.00%)	3 (11.11%)	2 (16.67%)	0 (0.00%)	1 (2.38%)	
Criterion	83 (86.45%)	4 (100.00%)	20 (74.07%)	10 (83.33%)	11 (100%)	38 (90.48%)	
Other	7 (7.29%)	0 (0.00%)	4 (14.81%)	0 (0.00%)	0 (0.00%)	3 (7.14%)	
** *Chi-Square (All)* **	χ^2^(2) = 121.94, *p* < 0.001						
**SM Use Measure**	***n* = 111**	***n *= ** **5**	***n *= ** **27**	***n *= ** **16**	***n *= ** **11**	***n *= ** **52**	χ^2^(4) = 4.26, *ns*
Predictor	89 (80.18%)	4 (80.00%)	21 (77.78%)	13 (82.25%)	9 (81.82%)	42 (80.77%)	
Criterion	16 (14.41%)	1 (20.00%)	3 (11.11%)	2 (12.50%)	1 (9.09%)	9 (17.31%)	
Other	6 (5.41%)	0 (0.00%)	3 (11.11%)	1 (6.25%)	1 (9.09%)	1 (1.92%)	
** *Chi-Square (All)* **	χ^2^(2) = 110.97, *p* < 0.001						
**Relationship Tested**	***n* = 162**	***n *= ** **8**	***n *= ** **38**	***n *= ** **20**	***n *= ** **11**	***n *= ** **85**	χ^2^(4) = 14.86, *p* = 0.005
No	75 (46.30%)	5 (62.50%)	10 (26.32%)	9 (45.00%)	2 (18.18%)	49 (57.65%)	
Yes	87 (53.70%)	3 (37.50%)	28 (73.68%)	11 (55.00%)	9 (81.82%)	36 (42.35%)	
** *Chi-Square (All)* **	χ^2^(1) = 0.89, *ns*						
**Relationship Nature**	***n* = 87**	***n *= ** **3**	***n *= ** **28**	***n *= ** **11**	***n *= ** **9**	***n *= ** **36**	χ^2^(4) = 3.02, *ns*
Negative	1 (1.15%)	0 (0.00%)	0 (0.00%)	0 (0.00%)	0 (0.00%)	1 (2.78%)	
No relationship	5 (5.75%)	0 (0.00%)	0 (0.00%)	1 (9.09%)	0 (0.00%)	4 (11.11%)	
Positive	81 (93.10%)	3 (100.00%)	28 (100%)	10 (90.91%)	9 (100.00%)	31 (86.11%)	
** *Chi-Square (All)* **	χ^2^(2) = 140.14, *p* < 0.001						

Notes. To test for differences across different study types, we used Fisher’s exact test, given that some cells included less than five cases.

## 4. Discussion

### 4.1. Overview of Findings

This SLR builds upon and extends three previous reviews of the literature on social media use and alcohol [14,15,16,17]. This review extends the scope of previous reviews in important ways by addressing three main aims: (1) to describe the prevalence, methods, and theories related to alcohol-related social media studies; (2) to describe the findings regarding the relationship between social media and alcohol use; and, (3) to compare studies as a function of the type of social media content and uses examined in the studies.

Regarding the first aim (see second column from the left in Table 2), our findings showed important trends within this research domain. First, research in this domain tends to be situated in peer-reviewed journals, mostly focused on the United States and other Western countries and split almost evenly in terms of funding. The research tends to rely heavily on quantitative methods (mostly survey-based), cross-sectional in nature, and is conducted mostly online. The research reviewed here tends to be situated within Anglo-Saxon research traditions that value positivist and post-positivist methods of knowing. The overrepresentation of Western geographic locations, coupled with heavier emphasis on quantitative methodologies, does call for a more critical assessment of the overarching research agenda in this domain. Additionally, the research reviewed here mostly focuses on adult populations and mixed-age samples, with few studies focusing on those below the age of 21, and a considerable percentage of studies (over four in 10) that do not specify the age group of study participants. This finding is significant in two distinctive ways. First, it is plausible that the difficulty of recruiting children and adolescents results in underrepresentation of younger populations. However, it is important to note that studying these populations is extremely important, given past research suggesting direct and long-term socio-health effects of early onset drinking [29], especially that these younger populations are also heavier adopters and users of social media [30]. Second, the lack of specificity in terms of the age of participants could be a function of the nature of data collection. Given that several studies leveraged big-data analyses of social media content and given platform restrictions in terms of collecting personal data, such information is missing, thus, rendering the question about impact on vulnerable populations unanswered.

In terms of the specifics of how alcohol consumption was operationalized in the reviewed studies, the studies tended to favor self-report measures (single study using unobtrusive measures, and over half did not specify the nature of the alcohol use measure), where alcohol use was not measured within a timeframe. Additionally, most studies did not leverage clinical and established measures of alcohol use (e.g., AUDIT), did not measure perceptions of drinking among different reference groups, including friends, and shied away from leveraging the social norms approach as an operational framework for social influence. The lack of uniformity in operationalizing alcohol use needs further attention by researchers in this field, especially that clinically acceptable measures could be necessary to further situate this this area of inquiry within the public health domain. Secondly, the lack of social contextualization of alcohol consumption, and especially given the ‘social’ nature of social media, is a missed opportunity for extending our understanding of this phenomenon beyond the individual focus by incorporating social and interpersonal relationships as factors driving both alcohol and social media uses. Similar concerns emerge where much of the research reviewed here disregards measures of excessive drinking, negative consequences and risks associated with drinking, concurrent alcohol use with other drugs, and celebration drinking. Such gaps in the literature should inform the future research agenda to expand the focus from the individual general use of alcohol to more socially informed aspects of alcohol consumption and overuse, along with other risky and regulated substances.

Our findings showed that Facebook maintained the lead in terms of the context of examining the phenomenon at hand, with platforms such as Twitter, Instagram, YouTube trailing behind. It is worth nothing that a considerable portion of the studies examined social media use as a general area of use without specifying any platform as the context of the study as well as a portion of the studies that examined multiple platforms concurrently. It is worth mentioning that newer social media platforms, such as TikTok, have not been represented in the sample of studies reviewed here and there was a low prevalence of studies on exponentially growing platforms such as Instagram and Snapchat, which was plausibly due to the recent increased popularity of such platforms, thus not leaving much time for the academic literature to catch up with these trends. One noteworthy finding is the low prevalence of studies measuring time spent on social media and the number of friends on social media as metrics to index social media use. It is apparent that social media use has been dealt with in a categorical way, outside the boundaries of more robust measures of extent of use and other important factors that index this use (i.e., number of friends).

One of our major findings relates to the severe lack of theory-based research in this domain. Less than one-third of the studies leveraged a theoretical framework, and these theories were dispersed across different disciplines. The profile of studies included in this SLR spans multiple disciplines, and in several cases, the research is published in interdisciplinary journals, yet much of the theories used tend to examine the phenomenon on the psychological level. This could be a plausible explanation for the lack of clear theoretical foundations of the research in this area. However, as this field matures, it is critical for more theory-based investigations that not only serve to predict aspects of human and technology-related behaviors but rather could aid with designing effective strategies and interventions to combat alcohol-related risks that are further elevated within the social and digital media environments. Such practical extensions of this body of literature could also be informed by greater emphasis on sociological and critical theoretical frameworks to examine the impact of social and cultural structures, values, and practices on the relationship between alcohol use and social media use.

To address the second aim in our SLR, we examined how the relationship between social media use and alcohol consumption was operationalized and tested. One of the most intriguing findings in our SLR deals with providing a clear determination that social media use is an antecedent to alcohol consumption within this body of literature. Despite the low frequency of experimental research in our sample, replicative evaluation of the associative relationship between the two constructs equips our team to propose a causal order for this relationship, albeit mostly reliant on associative examination of that relationship: social media use predicts alcohol consumption, and this relationship is squarely positive, in that greater social media use is associated with higher alcohol consumption.

Finally, to address this study’s third aim, we compared the corpus of studies as a function of the social media content and use (SMCU) types (see Table 2). We examined areas of commonality and divergence within the literature as it relates to these content categories. First, our findings showed that the different SMCU types were similar in terms of the distribution of publication venues, funding disclosures, research methods, types of qualitative methods, research design, distribution methods, inclusion of drinkers and non-drinkers, heavier use of self-report measures of alcohol use, lack of integration of social norms, questions about excessive drinking, measurement of perceived negative consequences and risks, focus on celebration drinking, and integrating network-related factors (e.g., number of friends). Additionally, across different SMCU types, there was harmony in terms of regarding alcohol consumption as a criterion to social media use, where the relationship tends to be positive. Taken together, these commonalities indicate harmony within the literature, yet they also point to important implications for future research. The harmonious operationalization of the relationship between the two constructs enhances the validity of assessing such a relationship, which provides a foundation for research that extends beyond the parameters of this operationalization by diversifying the platforms, affordances, and content types examined in future research. However, such harmony also points to uniform gaps in terms of contextualizing alcohol use using social and cultural lenses. More research could emphasize such gaps.

In terms of divergence, there are several important findings to discuss. First, the divergence in terms of the types of quantitative methods used in this domain is important to drive cross-pollination across disciplines. We see that advertising and marketing research in this area tends to be heavily focused on experimental work, yet other types favor survey-based research. It is important to reflect methodological diversity within both quantitative and qualitative approaches to better formulate a comprehensive understanding of this socio-technical/health phenomenon. Second, the divergence also points to a lack of cross-disciplinary standardization of how the two constructs were operationalized. Wider discussions within the field and across fields is necessary to further harmonize these methods of operationalization. Finally, our review points to methodological divisions in terms of examining the two constructs within the same study. Much of the research did not formally test the relationship, despite alluding to it. Whether conducting a quantitative or qualitative study, it is important for research in this domain to further showcase the extent to which the two constructs are related. This is also important regarding pushing the boundaries in what we termed here as big-data analyses, where researchers, with considerable mastery, mined and analyzed (through computer- and human-based coding) large data sets of social media posts related to alcohol. A few of the studies attempted to link social media content to community-related alcohol use and incidents of alcohol overuse (e.g., alcohol-related visits to emergency rooms). More of the latter type of research is needed within this domain, where the emphasis should evolve into highlighting the impacts of digital touchpoints on individuals and societies.

### 4.2. What Are the Gaps, What Don’t We Know, Where Should We Go?

To address this topic, we provide an overview of topics previous reviews called for in new research on social media and alcohol, and then within them, we offer insights garnered from the results of this systematic review.

First, previous reviews on social media and alcohol use have called for longitudinal studies that can establish the causal order of the variables [14,15,16], and, in particular, Lobstein et al. [15] highlight a need for longitudinal data identifying brand-specific youth exposure and brand-specific alcohol consumption to demonstrate more convincingly the causal links from marketing activity to drinking behavior. Our SLR arrived at a similar conclusion given the higher prevalence of cross-sectional rather than longitudinal research. Based on this, we mirror the previous reviews’ call for more longitudinal research in this field of study.

Second, previous reviews called for greater investment in theory-driven research in this area of research, the lack of which was evident in our review, thus intensifying the need for more theory-based research here. It is worth mentioning that our review highlighted a strong diversity of theoretical frameworks across academic disciplines that examined the relationship between social media, which we deem as a point of strength for this area of research. However, we argue that there is a need to, potentially, organize the theorizing in this area of research across epistemological approaches and greater distribution among different disciplines, especially that most theories used in the literatures were psychological in nature. Additionally, Groth et al. [14] noted that the social norms approach was the most common framework identified in their review (as we found here). Novel to this review, though, is our findings showed that most studies disregarded the short- and long-term risks associated with drinking, celebration drinking, and none mentioned protective behaviors that can mitigate extreme alcohol consumption [31]. Future research should evolve into an arena of looking at this phenomenon in a more longitudinal manner that takes into consideration the health impact of the relationship between social media use and alcohol consumption. This ties in well with the call for research on peer influence on the relationship between social media use and alcohol.

Third, similar to Groth et al. [14], our findings showed a hegemony of quantitative methodological approaches, and within those studies, a high prevalence of survey-based and content-specific (content analysis and big data mining) research, whereas qualitative and experimental research took the backseat in driving the research agenda in this field. Our recommendation for future research is to enhance the diversity of methodological approaches by increasing the number of experimental studies that further qualify the causal relationship between social media use and alcohol consumption as well as qualitative studies that provide in-depth, sociological, and anthropological interpretations of this phenomenon.

Fourth, Facebook still maintains the lead in terms of study context. While Facebook is still the leading social media platform use worldwide, recent increases in popularity, platform substitution, and prevalence of newer uses of social media platforms have emerged, specifically, among younger users. With that in mind, future research should reflect those dynamic changes in the social media scene and account for the changing rituals and habitual uses of these platforms. For example, Snapchat, which entailed a low prevalence of studies in our sample, has been increasingly used for sharing alcohol-related content and messages, specifically due to its affordances related to disappearing content after a short period of time, thus alleviating privacy concerns with sharing such type of risky content. Another emerging platform worth examining is TikTok, which has exponentially increased in popularity over the past few years. It is also worth mentioning that for certain types of studies, specifically content-analytic and big data studies, a higher prevalence was observed for platforms such as Twitter, which was plausible due to the ease of extracting data from such platforms in comparison to other platforms with high API restrictions (e.g., Facebook, Instagram, and TikTok). Finally, it is important to expand on how researchers in this field measure social media use to include more robust measures of time spent on social media and number of friends, including server-based accounts of social media use. Future research should also reflect the growth and evolution in terms of affordances and functions of different platforms to further decipher the most important pain points related to the impact of social media use on alcohol consumption.

Fifth, despite our efforts in the current SLR to include a wide range of databases, most of the studies were in English and centered in the United States and other Western countries. Alcohol use, overuse, and abuse is a global public health concern, which causes three million deaths annually, representing 5.3% of all deaths globally [32]. Academic journals and professional associations should further capitalize on their concerted efforts to expand the geographic reach of scholarly research in this arena, given its significant impact on the health and well-being of individuals around the world.

## 5. Conclusions

In summary, our coding of the literature by the types of content studies provided us with several intriguing operational and theoretical insights into the large body of research focusing on the use of social media and how that relates to alcohol use and overuse. Throughout our coding, we extrapolated differences between advertising and marketing research in how it reflected the relationship between alcohol advertising and marketing, thus shedding the light on persuasive branded communication. The two coding categories related to UGC stratified the focus into self-generated social media posts and other-generated posts, thus highlighting the interpersonal (or hyperpersonal, see [19]) nature of social media as platforms where users gather to maintain their interpersonal relationship, and in doing so generate content themselves and become exposed to and interact with content generated by others. The fourth category dealt with operationalizing social media use in terms of frequency of use (e.g., time) and platform affordances (e.g., network size, type of use, etc.). Finally, a fifth category—most of the studies included in this SLR —entailed a combination of the four categories in relation to content types and uses. This categorization showcases the complexity of social media as a phenomenon and as a social organism with boundaries and attributes resembling a biological organism in terms of having a defined structure, metabolizes, grows in size and complexity, yearns for homeostasis, responds to stimuli, reproduces, and adapts/evolves [33]. Much like this understanding of social media, we can see, is evident in our SLR, where the examination of the relationship between social media use and other aspects of human livelihood, i.e., health and well-being, commerce, economy, etc., is sensitive to how these social organisms evolve and grow. In essence, this SLR further explicates the complex relationship between the online and offline worlds. In short, a body of over 200 studies showed that this relationship is existent in our minds, culture, society, and in quantitative data samples at various levels starting from the individual, to groups, and countries. The relationship between use of a particular platform and risky behaviors, such as alcohol use and overuse, provides us with an understanding of the importance of a multi-faceted approach to curbing the potentially harmful effects of different aspects of dependence, be it on social media or alcohol.

## Figures and Tables

**Figure 1 ijerph-19-11796-f001:**
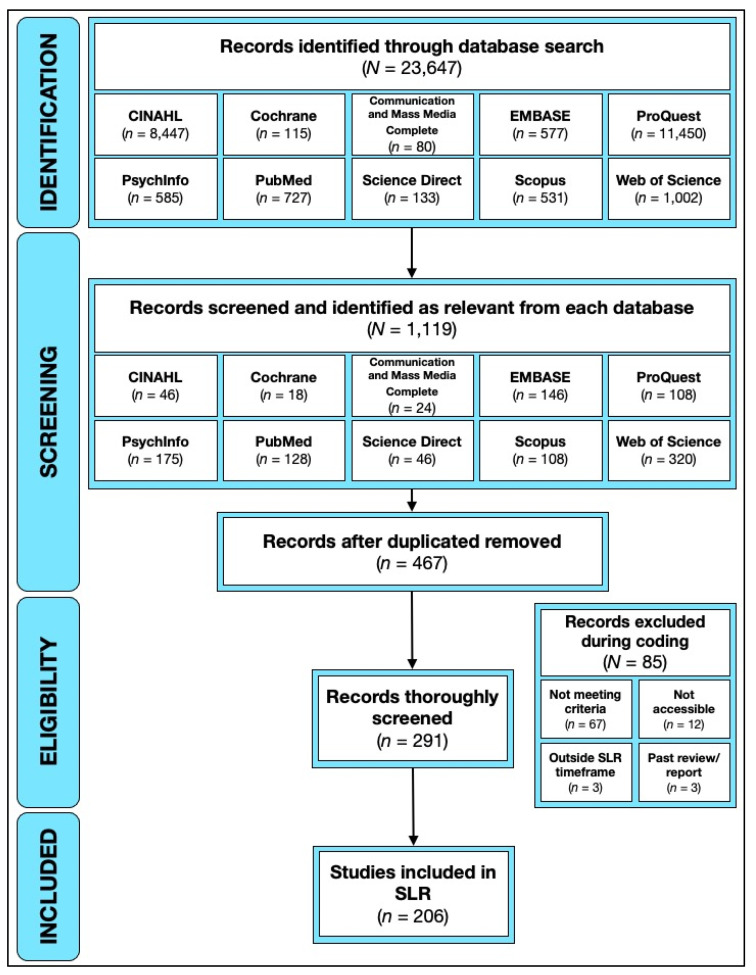
PRISMA Flow Diagram.

**Figure 2 ijerph-19-11796-f002:**
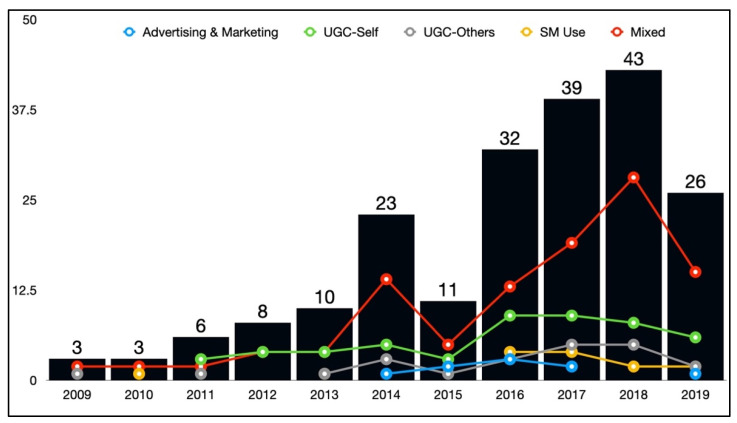
Frequency of alcohol-related social media studies, by year and study focus type.

**Figure 3 ijerph-19-11796-f003:**
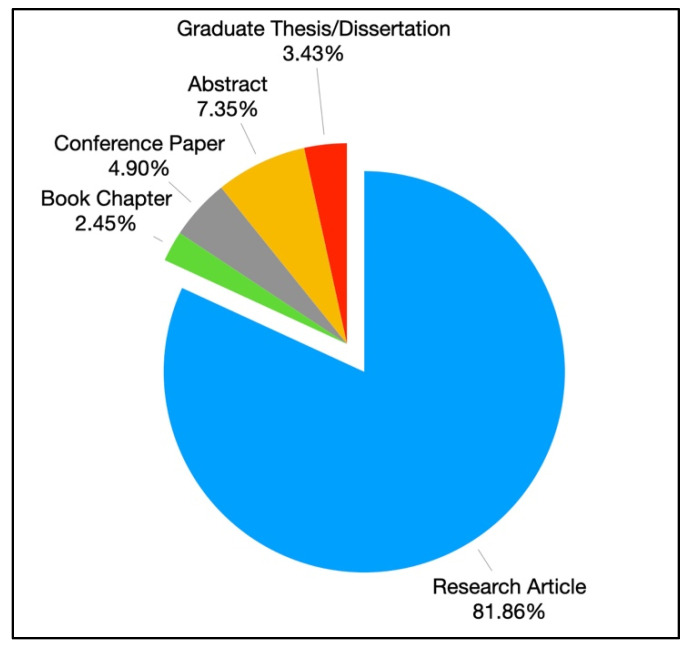
Frequency distribution of article types (Sum of percentages is not equal to 100% due to rounding).

**Figure 4 ijerph-19-11796-f004:**
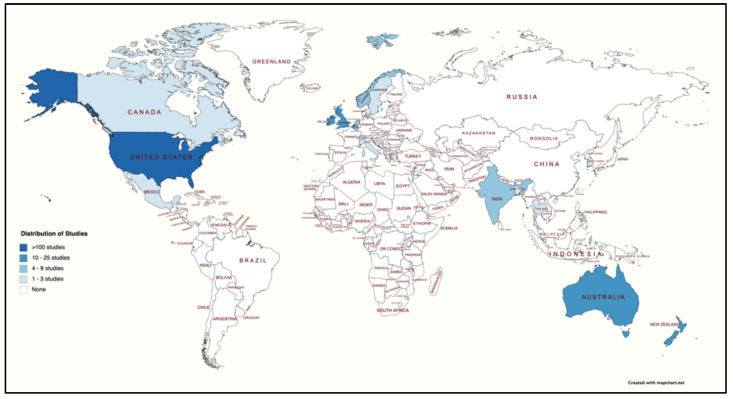
Distribution of studies included in the systematic review by country.

**Figure 5 ijerph-19-11796-f005:**
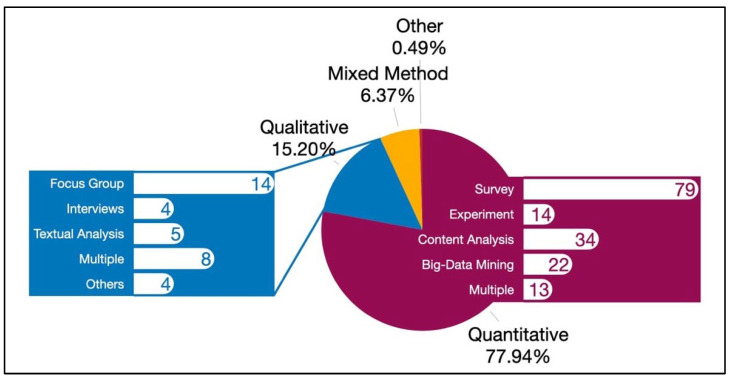
Frequency distribution of articles included in the systematic review, by research methodology.

**Figure 6 ijerph-19-11796-f006:**
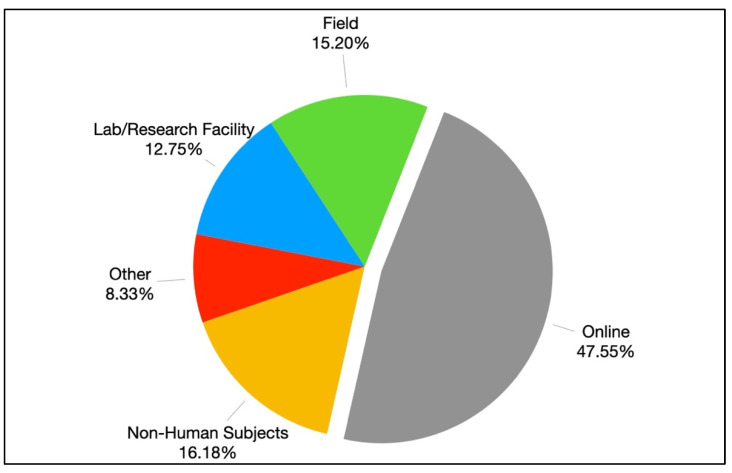
Distribution of research context.

**Figure 7 ijerph-19-11796-f007:**
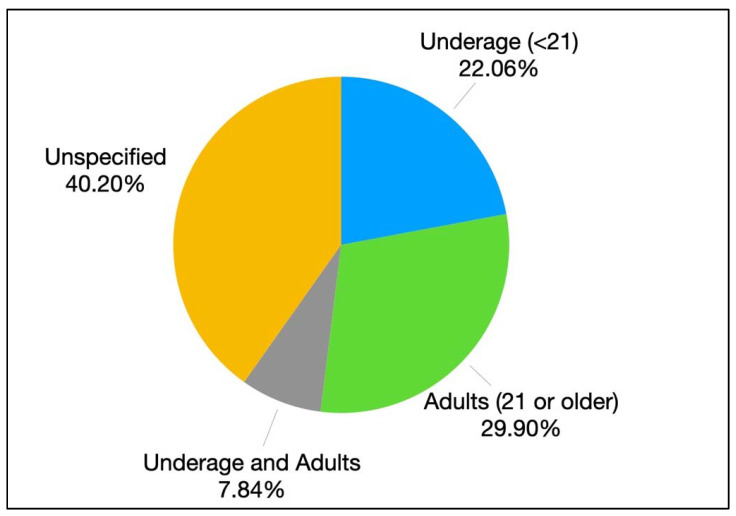
Distribution of studies included in systematic review by samples age group.

**Figure 8 ijerph-19-11796-f008:**
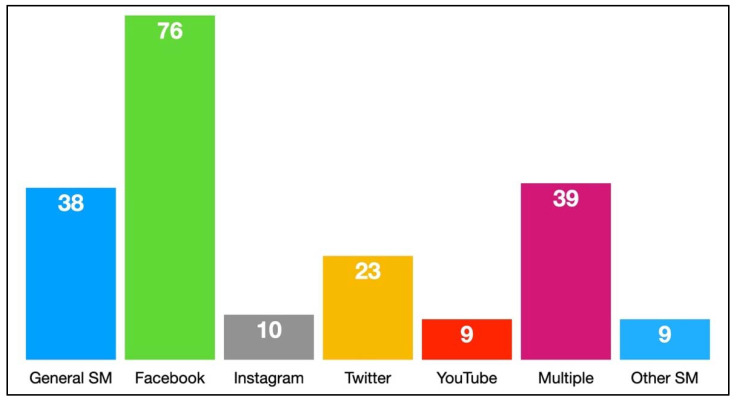
Distribution of studies included in the systematic review, by social media platform.

**Table 1 ijerph-19-11796-t001:** List of theories used in 67 of the 204 articles included in the analysis.

Psychological Perspective (31 Theories)	
Affordances Attribution Theory Co-construction Theory Elaboration Likelihood Model Expectancy-Value Models (*N* = 3) Facebook Influence Model (*N* = 3) Gestalt Theory Homophily Norms Learning Theory of Addiction Limited Capacity Model of Mediated Motivated Message Processing (LC4MP) Media Effects Model Media Practice Model Mere Exposure Effect Message Interpretation Process Priming	Psychological Reactance Theory Selective Exposure Theory (*N* = 3) Self-Concept Change Self-Perception Theory Self-Persuasion Self-Presentation (*N* = 2) Social Cognitive Theory (*N* = 4) Social Comparison Theory (*N* = 2) Social Identity Theory Social Information Processing Theory Social Learning Theory (*N* = 14) Social Network Theory Social Norms Approach (*N* = 22) Theory of Planned Behavior (*N* = 10) Theory of Reasoned Action (*N* = 9) Uses and Gratifications Theory (*N* = 2)
**Sociological (3 theories)**	**Critical (2 theories)**
Bourdieu’s Theory of Capitol Interactionist Theory of Sociology Media Convergence Theory	Feminist perspective Grounded theory (*N* = 3)
**Criminological (2 theories)**	**New Theory (1 theory)**
Deterrence Theory General Strain Theory	Pedagogy of Regret

Notes. All theories, except when noted with (*N*=), had a frequency of one; N refers to the frequency of times the theory is used across coded studies.

## Data Availability

Not applicable.

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
