# Peer review of "Social Media Use and Alcohol Consumption: A 10-Year Systematic Review"

_ijerph, 2022, doi:10.3390/ijerph191811796_

Round 1

Reviewer 1 Report

I think figure 1 is very illustrative, it helps to easily understand the PRISMA approach.

At the research design level, I am struck by the fact that, of the 206 studies, 87 are not focused on UGC or advertising. Perhaps other categories could be expanded, given that it accounts for 42%. More than any other category. 

Regarding the databases, I regret that no non-Anglo-Saxon research has been located, given that, despite being the most influential language, there is very relevant research in other Ibero-American databases that include research on this subject.

The data in Table 1 could be presented in a different way to facilitate understanding. It is not clear whether some of the theories are repeated in several articles. On the other hand, the 138 articles that do not declare themselves to a specific theory, inevitably position themselves from a certain academic perspective. What are they? Are they predominantly from communication sciences? From social psychology? From addiction pathology? From sociology?

The correlation between social networks and alcohol consumption can be better described. Being the central focus of the article, I feel that in the results block it is not excellently explained, which is a pity given that there is a huge amount of work behind it.

In the discussion I have the feeling that there are some redundancies in the presentation of results and perhaps more discussion is needed in relation to other similar studies.

In my view - obviously subjective - the conclusion should be broader, as opposed to the discussion which is too long. The conclusion should contextualise the work and show the findings with which it contributes to knowledge, so that the reader, who has to read numerous papers on a daily basis, can get an idea of the mission, scope and content of the research in hand.  

In short, without being a research that yields surprising findings, it does a good job in detecting overrepresented research formats and on the other hand points out the gaps that need to be filled in order to advance in this field of knowledge.

Author Response

Dear Reviewer 1,

We would like to, first, thank you for the careful review of our manuscript and the insightful feedback provided. In the following, we provide a summary of the changes to the manuscript in response to your comments.

I think figure 1 is very illustrative, it helps to easily understand the PRISMA approach.

Thank you very much for acknowledging the clarity brought by the PRISMA figure.

At the research design level, I am struck by the fact that, of the 206 studies, 87 are not focused on UGC or advertising. Perhaps other categories could be expanded, given that it accounts for 42%. More than any other category.

We would like to thank the reviewer for this observation. To address this comment, we re-analyzed the data and included alternative coding categories for the type of social media content included in the studies. Namely, we used the following social media content and uses (SMCU) types in the revised manuscript: advertising/marketing, self-user-generated content (UGC), other-UGC, social media uses and affordances, and mixed types of content. A broad description is presented in Section 1.2. These changes are reflected throughout the manuscript.

Regarding the databases, I regret that no non-Anglo-Saxon research has been located, given that, despite being the most influential language, there is very relevant research in other Ibero-American databases that include research on this subject.

We agree with the reviewer’s sentiment. In our approach, we attempted to expand our search process for locating literature to databases beyond those that restrict inclusion to research from Western and Anglo-Saxon countries. We will certainty expand on this issue in the Discussion section.

The data in Table 1 could be presented in a different way to facilitate understanding. It is not clear whether some of the theories are repeated in several articles. On the other hand, the 138 articles that do not declare themselves to a specific theory, inevitably position themselves from a certain academic perspective. What are they? Are they predominantly from communication sciences? From social psychology? From addiction pathology? From sociology?

Thank you for this insight. We have re-coded the theories used in the literature into major disciplines and elaborate on the implications of the overrepresentation of psychological approaches to examining the relationship between social media use and alcohol consumption. We have attempted to revisit the disciplinary focus of the publication’s venues of the different studies, yet, we were not able to validly identify clear categorization of journals, especially that studies in this domain are published across specialized disciplines (e.g., psychology, alcohol and other drugs, clinical journals, media and communication, and technology), yet several of the publications venues cut across these disciplines as they position themselves as interdisciplinary journals. We hope that this response is satisfactory to the address this reviewer’s comment. These changes are presented in the Results and Discussion sections.

The correlation between social networks and alcohol consumption can be better described. Being the central focus of the article, I feel that in the results block it is not excellently explained, which is a pity given that there is a huge amount of work behind it.

To address this comment, we enhanced the introduction section to address the importance of examining this relationship. We hope this change is satisfactory.

In the discussion I have the feeling that there are some redundancies in the presentation of results and perhaps more discussion is needed in relation to other similar studies.

Thank you for this insight. Per your recommendation, we have revised the Discussion section significantly. Instead of regurgitating the findings, we, instead, attempted to draw patterns to summarize our findings. We hope this revision is satisfactory.

In my view - obviously subjective - the conclusion should be broader, as opposed to the discussion which is too long. The conclusion should contextualise the work and show the findings with which it contributes to knowledge, so that the reader, who has to read numerous papers on a daily basis, can get an idea of the mission, scope and content of the research in hand. 

We revised the conclusion accordingly.

In short, without being a research that yields surprising findings, it does a good job in detecting overrepresented research formats and on the other hand points out the gaps that need to be filled in order to advance in this field of knowledge.

Thank you for the favorable opinion about the contribution of our study.

Thank you again for your effort in reviewing our manuscript and providing feedback that certainly further strengthened our manuscript.

Reviewer 2 Report

Recommendation: Major Revision

Comments:
Greetings, in my view by using a bibliometric approach the researchers have in essence aggregated the literature, rather than reviewed the literature. So, what begins as a cogent piece of work, ends with diffuse outcomes, so the paper loses focus. I would suggest that once the data has been aggregated, the review should focus on one element of Social Media Use and Alcohol Consumption or an interrelated set of elements of these two topics, thus providing a more meaningful discussion, outcomes and recommendations, it would also generate options for future research.  In addition, it appears that the researchers are not entirely comfortable with the subject matter, this is particularly evident in the introduction which is in essence a list of paraphrase's with no context, the same could be said of the implications. Also, there are issues with structure, layout, tables, figures headings, subheadings.

Additional Comments:
1. Originality: The paper has originality, however it seems to be less of a literature review, but rather an aggregation of literature in an attempt to identify key literature relevant to Social Media Use and Alcohol Consumption. The idea for the paper is original and can add value to this discipline, however my sense is that once the literature has been aggregated the paper loses focus as the subject area is too broad, therefore the focus continually changes throughout the paper. I would suggest that once the data has been aggregated, the review should focus on one element of Social Media Use and Alcohol Consumption or an interrelated set of elements, thus providing a more meaningful discussion, outcomes and recommendations, it would also generate options for future research.

2. Relationship to Literature:  There is a breath of literature, however due to the broad nature of the paper much of the literature is basic overview meaning the literature is laid out, but lacks interpretation which is a cornerstone of a good review. The introduction is a good example of this, it is in essence a list of quotations and paraphrase's that identify the key concepts, however there is little or no interpretation of the concepts that will enable the reader to generate the understanding required in order to address the subsequent methodology. There is also no attempt to identify the seminal literature in an outline of the key concepts, therefore much of the literature is orientated to the last 6 years or so (this is most likely an outcome of the aggregation of literature), I agree that this is a relatively recent phenomenon, however some attempt should have been made to identify seminal work.

3. Methodology:  Using a SLR approach is a reasonable approach, i have two comments, firstly, it is not clear as to why this is the most appropriate method to assess the literature (you need to convince your reader), this method also has its limitations, these should have been identified.

4. Results: the results were not presented in a cogent way, sub headings should be used, tables should be referred to within the document. Also due to the broad nature of the results, led to broad outcomes.

5. Implications for research, practice and/or society:  Yes there are implications, however they are diffuse, there is a requirement for a better approach to structure and layout of the implications, this would lead to better levels of clarity for the reader.

6. Quality of Communication: Over all the communication is fine, there are some typographical errors, also there are some issues with past and present tense.  There are issues with the table of journals, readers (including me) are not familiar with the acronyms. The introduction to the paper requires revision to provide clarity with regards to key terminology for the reader.

Author Response

Dear Reviewer 2,

We would like to, first, thank you for the careful review of our manuscript and the insightful feedback provided. In the following, we provide a summary of the changes to the manuscript in response to your comments.

Greetings, in my view by using a bibliometric approach the researchers have in essence aggregated the literature, rather than reviewed the literature. So, what begins as a cogent piece of work, ends with diffuse outcomes, so the paper loses focus. I would suggest that once the data has been aggregated, the review should focus on one element of Social Media Use and Alcohol Consumption or an interrelated set of elements of these two topics, thus providing a more meaningful discussion, outcomes and recommendations, it would also generate options for future research. 

Thank you for your comment and recommendation. We believe, and based on Reviewer 1’s recommendation, we have re-operationalized the coding variable of study content types. In doing so, we have streamlined points of commonality and divergence among these different categories.

In addition, it appears that the researchers are not entirely comfortable with the subject matter, this is particularly evident in the introduction which is in essence a list of paraphrase's with no context, the same could be said of the implications. Also, there are issues with structure, layout, tables, figures headings, subheadings.

We thank the reviewer for this feedback. We understand that the structure of a systematic literature review can be different from a traditional empirical study. In an effort to conserve space for the presentation and discussion of the results, we opted for a brief front end. We have, however, refined the front end to address some of the issues raised here.  

We also would like to thank the reviewer for the comment related to the “layout, tables, figures headings, subheadings.” We have followed the journal template in creating the structure and layout of this manuscript, and if we had made errors, we trust that the editorial team in the journal will carefully review the manuscript, in hope it would gets accepted, before publication.

Additional Comments:
1. Originality: The paper has originality, however it seems to be less of a literature review, but rather an aggregation of literature in an attempt to identify key literature relevant to Social Media Use and Alcohol Consumption. The idea for the paper is original and can add value to this discipline, however my sense is that once the literature has been aggregated the paper loses focus as the subject area is too broad, therefore the focus continually changes throughout the paper. I would suggest that once the data has been aggregated, the review should focus on one element of Social Media Use and Alcohol Consumption or an interrelated set of elements, thus providing a more meaningful discussion, outcomes and recommendations, it would also generate options for future research.

We thank the reviewer for this feedback. In conducting this laborious systematic literature review, our intent was to take a bird’s eye view of the overarching field of research linking social media use to alcohol consumption. We opted for a descriptive and narrative textual analysis with use of quantitative coding for our method of SLR. Given that other SLR and meta analyses have addressed more narrowly defined constructs within each of the aforementioned concepts, we felt that our unique contribution in this study is, actually, what seems to be disfavored by the reviewer, which is taking a more holistic, broad view of the field. We have elaborated on this in the first part of the Materials and Methods section of the revised manuscript.

  1. Relationship to Literature:  There is a breath of literature, however due to the broad nature of the paper much of the literature is basic overview meaning the literature is laid out, but lacks interpretation which is a cornerstone of a good review. The introduction is a good example of this, it is in essence a list of quotations and paraphrase's that identify the key concepts, however there is little or no interpretation of the concepts that will enable the reader to generate the understanding required in order to address the subsequent methodology. There is also no attempt to identify the seminal literature in an outline of the key concepts, therefore much of the literature is orientated to the last 6 years or so (this is most likely an outcome of the aggregation of literature), I agree that this is a relatively recent phenomenon, however some attempt should have been made to identify seminal work.

We thank the reviewer for this comment. We have refined the introduction and presentation of the concepts to be more interpretive rather than direct verbatim quotes from the literature. We hope this revision satisfies the reviewer.

  1. Methodology:  Using a SLR approach is a reasonable approach, i have two comments, firstly, it is not clear as to why this is the most appropriate method to assess the literature (you need to convince your reader), this method also has its limitations, these should have been identified.

Thank you for this note. We added a paragraph at the beginning of the Materials and Methods section to address this comment.

  1. Results: the results were not presented in a cogent way, sub headings should be used, tables should be referred to within the document. Also due to the broad nature of the results, led to broad outcomes.

Per this reviewer and the other reviewers’ recommendation, we have re-analyzed the data and attempted at presenting them in clearer ways. We attempted to organize our results section in accordance with the outlined research questions. Additionally, our results section was marked by headings and sub-headings and we ensure all tables and figures were referenced in the body of the results.

  1. Implications for research, practice and/or society:  Yes there are implications, however they are diffuse, there is a requirement for a better approach to structure and layout of the implications, this would lead to better levels of clarity for the reader.

We have refined the Discussion section to offer a set of clearer implications for theory and practice.

  1. Quality of Communication: Over all the communication is fine, there are some typographical errors, also there are some issues with past and present tense.  There are issues with the table of journals, readers (including me) are not familiar with the acronyms. The introduction to the paper requires revision to provide clarity with regards to key terminology for the reader.

Thank you for the comment. We have revised the manuscript accordingly and fixed the errors throughout the manuscript.

Thank you again for your effort in reviewing our manuscript and providing feedback that certainly further strengthened our manuscript.

Reviewer 3 Report

The aim of the paper is very clear from the abstract phase. The title is relevant and informative.

The introduction is clear and the research question is very clearly outlined. However, I was surprised to find that there are only two systematic literature reviews and one meta-analysis related to social media and alcohol use. This part of my review took most of the time to document. I found only Vannucci, A., Simpson, E. G., Gagnon, S., & Ohannessian, C. M. (2020). Social media use and risky behaviors in adolescents: A meta-analysis. Journal of Adolescence79, 258-274, but this meta-analysis included other risky behaviors, not only alcohol consumption.

I suggest authors to detail the process of arriving at 1119 records screened and identified as relevant from 23,647 records identified through database search. At line 240, I suppose that ”students” should be replaced with ”studies”.  I also suggest authors to detail the criteria for choosing 291 records thouroghly screened out of 467 records after duplicated removed.

The Table 1 (List of Theories) is highly appreciated.

I suggest authors to include a summary of studies included in this literature review. Maybe by adding a column at Table 1 and referring 2-3 relevant papers for each theory.

The overall merit of article is clear and deserved to be published in the present form, but the above suggestions may increase its impact after publication.

Author Response

Dear Reviewer 3,

We would like to, first, thank you for the careful review of our manuscript and the insightful feedback provided. In the following, we provide a summary of the changes to the manuscript in response to your comments.

The aim of the paper is very clear from the abstract phase. The title is relevant and informative. 

Thank you!

The introduction is clear and the research question is very clearly outlined. However, I was surprised to find that there are only two systematic literature reviews and one meta-analysis related to social media and alcohol use. This part of my review took most of the time to document. I found only Vannucci, A., Simpson, E. G., Gagnon, S., & Ohannessian, C. M. (2020). Social media use and risky behaviors in adolescents: A meta-analysis. Journal of Adolescence79, 258-274, but this meta-analysis included other risky behaviors, not only alcohol consumption.

We thank the reviewer for this comment. Although the review was published outside of our study’s timeframe, we referenced it in the introduction. Thank you for guiding us in this regard.

I suggest authors to detail the process of arriving at 1119 records screened and identified as relevant from 23,647 records identified through database search. At line 240, I suppose that ”students” should be replaced with ”studies”.  I also suggest authors to detail the criteria for choosing 291 records thouroghly screened out of 467 records after duplicated removed.

We added more information related to our inclusion criteria and the process of narrowing down our search procedure.

The Table 1 (List of Theories) is highly appreciated. I suggest authors to include a summary of studies included in this literature review. Maybe by adding a column at Table 1 and referring 2-3 relevant papers for each theory. 

Thank you. Based on other reviewer’s recommendations, we further coded the theories into disciplines. We opted not to provide example papers for each theory, given that majority of the theories had a frequency of one (1) in the coded literature. Additionally, for certain theoretical approaches, such as the social norms approach, there is a higher frequency of 22 studies, thus including a sub-sample would be hard to choose, but would also change the nature of the descriptive SLR we conducted.

The overall merit of article is clear and deserved to be published in the present form, but the above suggestions may increase its impact after publication.

Thank you very much for this favourable assessment of our manuscript.

Thank you again for your effort in reviewing our manuscript and providing feedback that certainly further strengthened our manuscript.

Round 2

Reviewer 2 Report

Dear author(s),

I think the paper is more focussed and it presents stronger contributions and conclusions.

I can see that you followed all my suggestions.

Best Regards